# 5-Hydroxytryptophan, a Precursor for Serotonin Synthesis, Alleviated Cognitive Dysfunction in a Mouse Model of Sepsis-Associated Encephalopathy

**DOI:** 10.3390/biomedicines13102319

**Published:** 2025-09-23

**Authors:** Chen Zhang, Jianing Jiang, Yiran Zhang, Zheren Tan

**Affiliations:** 1Department of Pediatrics, Xiangya Hospital, Central South University, 87 Xiangya Road, Changsha 410008, China; xyyyzhangchen@163.com; 2Xiangya School of Medicine, Central South University, 87 Xiangya Road, Changsha 410008, China; xyyyjjn@163.com (J.J.); xyyyzhangyiran@163.com (Y.Z.); 3Department of Critical Care Medicine, Xiangya Hospital, Central South University, 87 Xiangya Road, Changsha 410008, China; 4Bioinformatics Center & National Clinical Research Centre for Geriatric Disorders & Department of Geriatrics, Xiangya Hospital, Central South University, Changsha 410008, China; 5Hunan Provincial Clinical Research Center for Critical Care Medicine, Changsha 410008, China

**Keywords:** cognitive dysfunction, serotonin, sepsis-associated encephalopathy, serotonergic neurotransmission, therapeutics

## Abstract

**Background**: Patients with sepsis-associated encephalopathy (SAE) present with cognitive impairments. Serotonergic neurotransmission plays a critical role in regulating cognitive processes, and its dysfunction may contribute to SAE-related deficits. However, the effect of 5-hydroxytryptophan (5-HTP), a direct serotonin precursor, on SAE has not been investigated. We hypothesized that 5-HTP could alleviate cognitive dysfunction in SAE. **Methods**: The SAE mouse model was induced via intraperitoneal administration of lipopolysaccharide (LPS, 10 mg/kg). Cognitive function and locomotor activity were assessed using the Barnes maze, novel object recognition test, and open-field test to evaluate the effects of 5-hydroxytryptophan (5-HTP). Additionally, WAY100635, a selective 5-HT1A receptor antagonist, was co-administered with 5-HTP to investigate the potential mechanisms underlying its effects on SAE-related cognitive dysfunction. The effects of 5-HTP and WAY100635 on cognition and motor activity were also investigated in healthy mice. **Results**: LPS-induced sepsis caused a learning deficit. A dose of 10 mg/kg 5-HTP improved cognitive dysfunction, whereas doses of 25 and 100 mg/kg worsened cognitive dysfunction. Moreover, 100 mg/kg 5-HTP increased mortality in SAE mouse models. Neither 5-HTP (10 mg/kg) nor WAY100635 (1 mg/kg) alone exerted a significant impact on the locomotor activity or cognitive function of healthy mice. The cognition-enhancing effect of 5-HTP (10 mg/kg) was reversed by WAY100635 (1 mg/kg). **Conclusions**: improvement in cognitive dysfunction by 5-HTP suggests that serotonergic transmission plays a role in the pathophysiology of SAE, and 5-HTP, an over-the-counter supplement approved for human use, may hold clinical potential for the prevention and treatment of SAE.

## 1. Introduction

Sepsis-associated encephalopathy (SAE) is a primary cause of brain disorders in global ICUs [1]. It is characterized by widespread brain dysfunction due to sepsis, without direct CNS infection, structural lesions, or other encephalopathy types [1]. Approximately 70% of patients with sepsis develop SAE [2,3], which is marked by cognitive deficits, memory issues, and behavioral changes [4]. Evidence shows SAE significantly raises in-hospital mortality and contributes to adverse clinical outcomes in septic patients [5]. Consequently, up to 45% of sepsis survivors suffer long-term cognitive decline, greatly affecting their quality of life [6].

Our previous study has demonstrated that serotonergic neurotransmission contributes to cognitive dysfunction in two well-established mouse models of SAE [7]. We observed reduced 5-HT levels in the hippocampus, brainstem, and frontal cortex of septic mice. Notably, administration of fluoxetine, a selective serotonin reuptake inhibitor (SSRI), effectively ameliorated cognitive impairments in these models, suggesting enhanced 5-HT neurotransmission may be key for SAE improvement.

However, such agents modulate not only serotonergic signaling but also other neurotransmitter systems, including dopaminergic, noradrenergic, and GABAergic pathways [8], and such off-target effects of 5-HT agents may potentially mask their therapeutic effects on cognitive dysfunction of SAE.

5-hydroxytryptophan (5-HTP), the precursor of 5-HT, is synthesized from L-tryptophan in the brain by tryptophan hydroxylase-2 and then converted to 5-HT by aromatic L-amino acid decarboxylase. Administration of 5-HTP can increase the synthesis and release of 5-HT in the brain [9]. Like SSRIs, 5-HTP is employed clinically in the management of various psychiatric conditions and is available as an over-the-counter oral supplement with a relatively favorable side effect profile. When enhancing central 5-HT levels, 5-HTP might be more advantageous than SSRIs due to its greater specificity, as SSRIs might cause nonspecific off-target effects as well as several undesirable side effects. However, the role of 5-HTP in SAE has not yet been investigated. Therefore, we hypothesized that 5-HTP can alleviate cognitive dysfunction of SAE.

In the present study, we further investigate the contribution of 5-HT neurotransmission to the cognitive impairments observed in SAE. We initially explored the dose–effect relationship of 5-HTP on cognitive performance in a mouse model of SAE and further investigated potential 5-HT receptor mechanisms underlying the cognition-enhancing effect of 5-HTP.

## 2. Materials and Methods

### 2.1. Animals

Male wild-type C57BL/6 mice (initial body weight 18–22 g, aged 8–10 weeks) were obtained from Hunan SJA Laboratory Animal Co., Ltd. (Changsha, China). Animals were group-housed (up to five per cage) under controlled environmental conditions (temperature 18–25 °C; relative humidity 50–60%; 12 h light/dark cycle) and provided with standard rodent chow and water ad libitum. To facilitate adaptation prior to sepsis-associated encephalopathy (SAE) model induction, mice were handled daily for 2 min over a 7-day period. All experimental procedures were conducted in accordance with ethical guidelines for animal welfare. Study design and all animal experimental protocols were approved by the Laboratory Animal Welfare and Ethics Committee of Central South University, Changsha, China (Protocol Number: CSU-2022-0414).

### 2.2. Drugs

Lipopolysaccharide (LPS) was purchased from Sigma-Aldrich (St. Louis, MO, USA). 5-Hydroxytryptophan (5-HTP) was obtained from MeilunBio (Dalian, China). WAY100635 was sourced from MedChemExpress LLC (Monmouth Junction, NJ, USA). LPS, 5-HTP, and WAY100635 were each dissolved in 0.9% saline (vehicle) and administered via intraperitoneal (i.p.) injection.

### 2.3. Experimental Design and Drug Administration

In the present study, male mice were randomly allocated into six experimental groups: Control, LPS, LPS + 5-HTP, Control + 5-HTP, Control + WAY100635, LPS + 5-HTP + WAY100635. SAE was induced by intraperitoneal (i.p.) administration of LPS at a dose of 10 mg/kg. Mice were administered 5-HTP (10, 25, or 100 mg/kg) or vehicle via intraperitoneal injection, 30 min prior to behavioral testing. In the Control + WAY100635 group, WAY100635 (1 mg/kg) was injected i.p. 30 min before the trial. In the LPS + 5-HTP + WAY100635 group, WAY100635 (1 mg/kg) was administered i.p. 30 min prior to 5-HTP injection. Following the final behavioral assessment, mice were anesthetized with isoflurane and euthanized by decapitation.

### 2.4. Behavioral Tests

All behavioral assessments were conducted between 09:00 and 17:00. Behavioral performances were recorded and subsequently analyzed, as previously described [7]. The schedule and sample sizes of behavioral tests are shown in Figure 1 and Table 1.

#### 2.4.1. Open-Field Test

The open-field test (OFT) is a useful and simple test first described by Hall [10] that assesses locomotor activity. OFT was conducted using a 40 × 40 × 50 cm plastic chamber. Mice were positioned at the center and permitted to explore freely for 10 min. Following each trial, the chamber was thoroughly cleaned with 70% ethanol and dried with paper towels to eliminate olfactory cues. Locomotor activity was quantified by measuring the mean velocity.

#### 2.4.2. Barnes Maze

The Barnes maze (BM) was employed to assess spatial learning and memory, as described previously [11]. The apparatus consisted of a circular platform (92 cm in diameter, elevated 105 cm above the floor) with 20 equally spaced holes positioned around the perimeter—19 of which were empty, and one led to an escape box. At the beginning of each trial, mice were placed in a cylindrical starting chamber at the center of the maze for 5 s. During the acquisition phase (learning phase), mice were allowed to explore the maze for up to 3 min per trial, with three trials conducted daily over five consecutive days. Mice that failed to locate the target hole within the allotted time were gently guided to the escape box and allowed to remain there for 1 min. Inter-trial intervals ranged from 30 to 45 min. In the probe trial (memory phase), the escape box was removed, and mice were permitted to explore the maze freely for 3 min. After each trial, the maze was cleaned with 75% ethanol and rotated to prevent the use of spatial cues. During the learning phase, primary latency (time to first encounter the escape box) was recorded and analyzed. During the memory trial, primary latency to the target hole and time spent in the target quadrant were recorded and analyzed.

#### 2.4.3. New Object Recognition Test

The novel object recognition test (NORT) was conducted to evaluate recognition memory, as described previously [7]. Mice were habituated to an empty open-field box (35 × 35 × 20 cm) for 10 min, 24 h prior to the acquisition phase. During the acquisition trial, mice were exposed to two identical objects and allowed to explore freely for 5 min. After a 2 h interval, a 5 min retention trial was performed to evaluate short-term memory by replacing one familiar object with a novel one. The discrimination index was calculated as follows: (T Novel − T Familiar)/(T Novel + T Familiar). Arenas and objects were cleaned with 70% alcohol between trials to minimize odor cues.

### 2.5. Statistical Analyses

All data are presented as mean ± standard deviation. Normality of data distribution was assessed using the Shapiro–Wilk and Kolmogorov–Smirnov tests, and homogeneity of variances was evaluated using Levene’s test prior to parametric analysis. When assumptions for parametric testing were not met, non-parametric alternatives were applied. Group comparisons were performed using the two-sample t-test for parametric data and the Mann–Whitney U test for non-parametric data. The results were considered statistically significant when *p* < 0.05. All statistical analyses were two-sided and were conducted using the Statistical Package for the Social Sciences version 23.0. GraphPad Prism 8 was used to make graphs.

## 3. Results

### 3.1. 5-HTP Improved Cognitive Dysfunction of LPS-Induced Septic Mice

The BM was employed to assess spatial learning and memory. Compared to the control group, the LPS group demonstrated significantly longer primary latency on days 2 and 3 of the learning phase (*p* = 0.006 and 0.029, respectively; Figure 2A). However, in the probe trial (memory phase), with the escape box removed, there were no significant differences in latency as well as time spent in the target quadrant between the LPS and control groups (Figure 2B,C). These results indicate that LPS (10 mg/kg) induces spatial learning impairment in mice.

The dose–response relationship of 5-HTP (10.25 or 100 mg/kg) on spatial learning and memory was assessed in the BM. During training phase, compared to the LPS group, 10 mg/kg of 5-HTP significantly shortened primary latency on days 3–5 (*p* = 0.027, 0.01, and 0.013, respectively); 25 mg/kg of 5-HTP prolonged primary latency on day 1 (*p* = 0.004); 100 mg/kg of 5-HTP increased primary latency on days 4 and 5 (*p* = 0.028 and 0.014, respectively; Figure 2A).

Furthermore, during the BM procedure, administration of 100 mg/kg of 5-HTP significantly decreased survival rate in septic mice (*p* = 0.009, Figure 2D). Given that the 100 mg/kg of 5-HTP dose exacerbated cognitive impairment and decreased survival rate in septic mice, this dose was not evaluated in subsequent behavioral experiments.

The NORT revealed no significant differences among groups in the discrimination index (Figure 2E). The open-field test (OFT) was used to assess locomotor activity, and no significant differences in mean velocity were observed between groups (Figure 2F), indicating that treatment with 5-HTP at doses of 10 and 25 mg/kg did not significantly affect locomotor activity in LPS-induced septic mice.

### 3.2. Effects of 5-HTP and WAY100635 on Cognitive and Motor Functions in Healthy Mice

Our previous study found that WAY100635 (5-HT1A receptor antagonist, 1 mg/kg) blocked the cognitive-enhancing effects of fluoxetine (SSRI) [7]. In order to establish the specificity of 5-HTP (10 mg/kg) and WAY100635 (1 mg/kg) in modulating cognitive impairment in SAE, we first characterized their effects in healthy mice. No statistically significant differences were observed among the Control, Control + 5-HTP, and Control + WAY100635 groups in performance on the BM, NORT, or OFT (Figure 3). These findings indicate that administration of 5-HTP (10 mg/kg) and WAY100635 (1 mg/kg) does not significantly alter cognitive or motor functions in healthy mice at the doses tested.

### 3.3. Cognition-Enhancing Effect of 5-HTP Was Blocked by WAY100635

To further explore the potential 5-HT receptor mechanisms underlying cognition-enhancing effect of 5-HTP, WAY100635 was administered 30 min prior to 5-HTP injection.

In comparison to the control group, the LPS group exhibited significantly longer primary latency on days 3,4 during learning phase of BM (*p* = 0.015 and 0.02, respectively, Figure 4A). The LPS + 5-HTP group demonstrated shorter primary latency on days 3–5 (*p* < 0.001, *p* = 0.01 and 0.004, respectively, Figure 4A) during learning phase and longer time spent in the target quadrant in the probe trial (*p* = 0.001, Figure 4C) of the BM compared to LPS group.

In comparison to the 5-HTP treatment alone, pretreatment with WAY100635 (1 mg/kg) prior to 5-HTP administration induced significantly prolonged primary latency on days 2–5 during the learning phase (*p* = 0.048, 0.003, 0.028, and 0.004, respectively, Figure 4A), and reduced time spent in the target quadrant in the probe trial of the BM (*p* = 0.006, Figure 4C). These findings demonstrate that WAY100635 (1 mg/kg) specifically antagonizes the cognitive-enhancing effects of 5-HTP in SAE. No significant differences were observed in cognitive performance or locomotor activity in NORT and OFT between groups.

## 4. Discussion

This study builds upon previous research indicating that serotonergic neurotransmission plays a crucial role in the cognitive impairments of SAE [7]. Our previous study demonstrated that fluoxetine, a SSRI, can ameliorate cognitive deficits in mouse models of SAE. The cognitive enhancing effects of fluoxetine were found to be mediated through the 5-HT1A receptor, as pretreatment with 5-HT1A receptor antagonist WAY100635 blocked its cognitive benefits.

Consistent with these findings, systemic administration of 5-HTP has been well documented to enhance the synthesis of 5-HT in the brain [9] and improved cognitive dysfunction of SAE mice in present study. Batch-to-batch variability is a well-recognized challenge in behavioral research [12], as subtle factors such as microbiome composition, husbandry conditions, or seasonal changes can influence cognition-related outcomes. This may explain why 5-HTP (10 mg/kg) consistently improved learning performance of SAE mice in both experimental sets, but enhanced probe-trial memory retention only in the latter.

This variability also highlights a broader challenge in preclinical behavioral studies: strict standardization reduces noise but may limit reproducibility, whereas controlled heterogenization (e.g., varying cage enrichment, age, or time of day) can enhance reliability across settings [13]. Subtle environmental differences between our two batches may have unintentionally introduced such heterogeneity, potentially contributing to the additional memory benefits observed in the second cohort of SAE mice.

Despite these differences, the consistent enhancement of learning underscores a robust cognitive effect of 5-HTP, while the additional probe-trial improvement further supports its potential benefits in SAE.

In addition, our study demonstrates that 5-HTP (10 mg/kg), administered at a dose not affecting cognitive function or locomotor activity in healthy mice, ameliorates cognitive deficits in the SAE mouse model. This cognitive-enhancing effect of 5-HTP was selectively blocked by the 5-HT1A receptor antagonist WAY100635, indicating that 5-HTP’s beneficial action against SAE-induced cognitive impairment is likely mediated specifically through 5-HT1A receptor signaling. These findings further suggest that the 5-HT1A receptor serves as a common pathway for both fluoxetine and 5-HTP.

However, our findings also highlight a critical consideration regarding the degree of serotonergic enhancement. While moderate doses of 5-HTP improved cognition, high doses (100 mg/kg) significantly increased mortality in septic mice, which suggests a narrow therapeutic window for 5-HTP administration. This paradoxical effect may stem from excessive serotonergic activity worsening systemic inflammation and organ dysfunction in sepsis. Indeed, 5-HT has been identified as a key mediator in inflammation [14]. Jingyao et al. found that 5-HT-sufficient mice had a significantly lower survival rate, elevated levels of TNF-α, IL-6, higher bacterial loads and more severe organ dysfunction (lung, liver, kidney, heart, etc.), compared to serotonin-deficient mice in CLP-induced sepsis [15]. An overload of peripheral 5-HT via high-dose 5-HTP likely mirrors these deleterious processes, exacerbating sepsis pathophysiology.

Furthermore, excess 5-HTP can precipitate serotonin syndrome characterized by neuromuscular hyperactivity, autonomic dysfunction, and hyperthermia [16], which has been observed in dogs treated with clinically relevant ingestion of 5-HTP [17]. The added burden of autonomic imbalance (e.g., temperature dysregulation, cardiovascular instability) may further exacerbate mortality in septic conditions. Thus, it is essential to emphasize the need for optimizing the dosage of systemically administered serotonergic agents in potential therapeutic applications.

## 5. Conclusions

Our findings demonstrate that 5-HTP improves cognitive dysfunction in a mouse model of SAE at an optimal dose, while high doses may worsen deficits and increase mortality. Behavioral evidence further suggests a possible involvement of the 5-HT1A receptor in mediating these effects. These results highlight the potential of 5-HTP as a candidate for further investigation in SAE but underscore the need for additional studies to clarify its underlying mechanisms and establish safety profiles for clinical application.

## Figures and Tables

**Figure 1 biomedicines-13-02319-f001:**
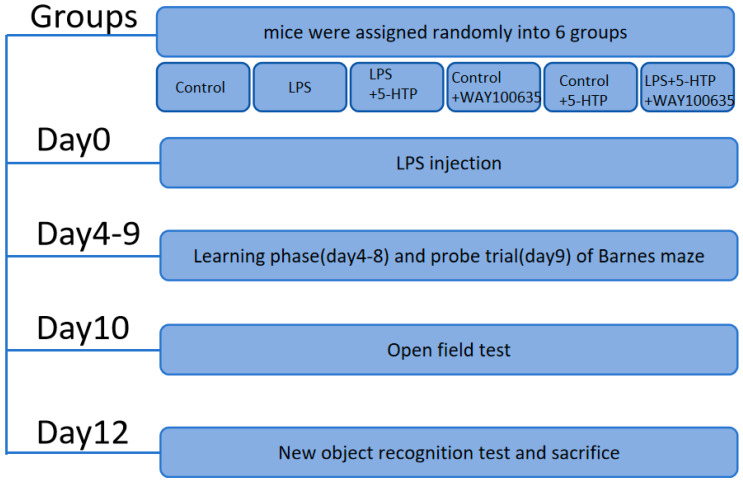
Flow chart of experimental design.

**Figure 2 biomedicines-13-02319-f002:**
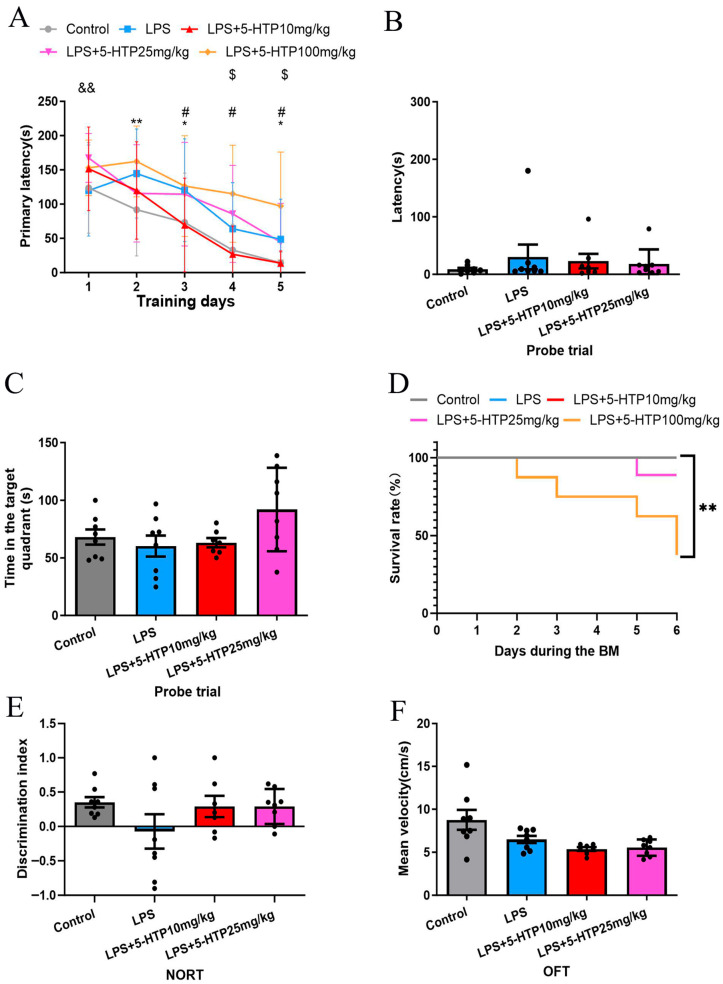
Effect of 5-HTP on cognitive behavior and motor functions in LPS-induced SAE model. (**A**) Performance during learning pahse of BM between groups; * Control versus LPS; # LPS versus LPS + 5-HTP (10 mg/kg); & LPS versus LPS + 5-HTP (25 mg/kg); $ LPS versus LPS + 5-HTP (100 mg/kg). (**B**,**C**) Probe trial of BM across groups. (**D**) Survival rate during BM procedure. (**E**) NORT discrimination index across group. (**F**) Mean velocity in OFT across groups. * *p* < 0.05, ** *p* < 0.01, # *p* < 0.05, && *p* < 0.01, $ *p* < 0.05. BM, Barnes maze; NORT, New object recognition test; OFT, Open-field test.

**Figure 3 biomedicines-13-02319-f003:**
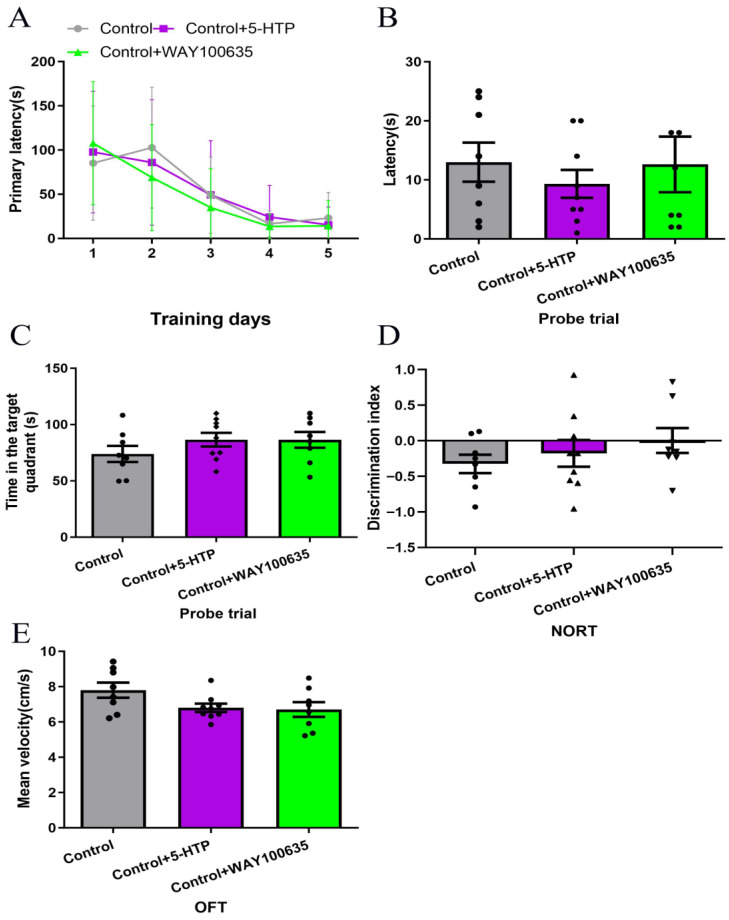
Effects of 5-HTP (10 mg/kg) and WAY100635 (1 mg/kg) on cognitive behavior and motor functions in healthy mice. (**A**–**C**) The BM performance (learning and probe trials) across groups. (**D**) NORT discrimination index across group. (**E**) Mean velocity in OFT across groups. BM, Barnes maze; NORT, New object recognition test; OFT, Open-field test.

**Figure 4 biomedicines-13-02319-f004:**
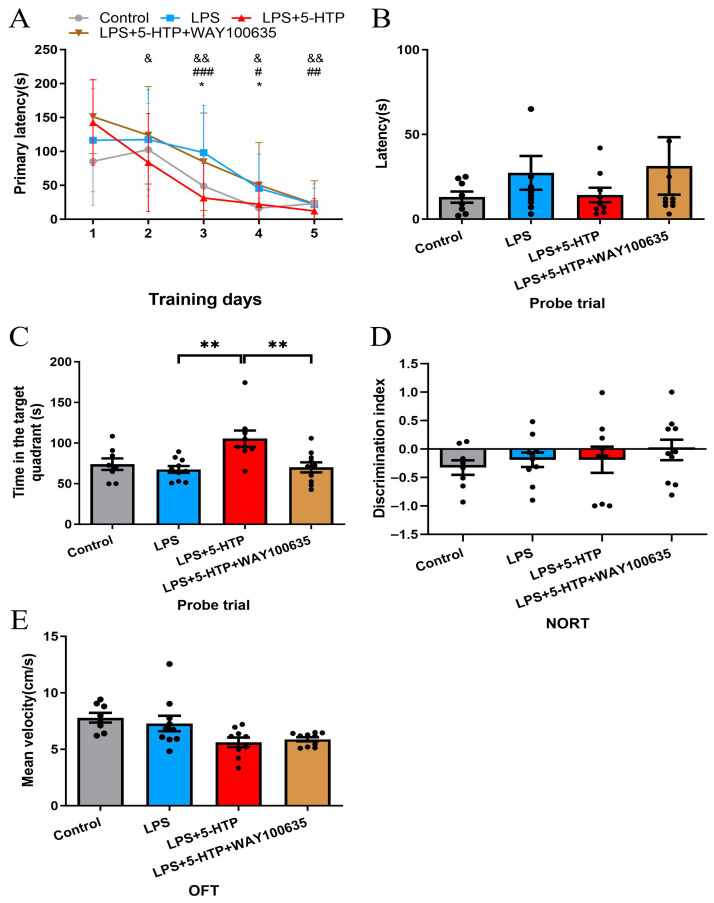
Impact of WAY100635 pretreatment on 5-HTP’s effects in LPS-induced SAE model. (**A**) Performance during learning phase of BM between groups; * Control versus LPS; # LPS versus LPS + 5-HTP; & LPS + 5-HTP versus LPS + 5-HTP + WAY100635. (**B**,**C**) Probe trial of BM across groups. (**D**) NORT discrimination index across group. (**E**) Mean velocity in OFT across groups. * *p* < 0.05, ** *p* < 0.01, # *p* < 0.05, ## *p* < 0.01, ### *p* < 0.001, & *p* < 0.05, && *p* < 0.01. BM, Barnes maze; NORT, New object recognition test; OFT, Open-field test.

**Table 1 biomedicines-13-02319-t001:** Summary of experimental groups, sample size and survival of behavior tests. BM, Barnes maze; NORT, New object recognition test; OFT, Open-field test.

Figure	Group	Type of Behavior Test	Sample Size of Group	Survival During Behavior Tests
Figure 2	Control	BM + NORT + OFT	8	All survived
	LPS	BM + NORT + OFT	8	All survived
	LPS + 5-HTP 10 mg/kg	BM + NORT + OFT	7	All survived
	LPS + 5-HTP 25 mg/kg	BM + NORT + OFT	8	Day 5 of BM:1 death
	LPS + 5-HTP 100 mg/kg	Learning phase of BM Probe trial of BM	6 3	Day 2, 3, 5 of BM:1 death per day Day 6 of BM:2 death
Figure 3 and Figure 4	Control	BM + NORT + OFT	8	All survived
	Control + WAY100635 1 mg/kg	BM + NORT + OFT	8	All survived
	Control + 5-HTP 10 mg/kg	BM + NORT + OFT	9	All survived
	LPS	BM + NORT + OFT	10	All survived
	LPS + 5-HTP 10 mg/kg	BM + NORT + OFT	9	All survived
	LPS + 5-HTP + WAY100635	BM + NORT + OFT	10	All survived

## Data Availability

The raw data supporting the conclusions of this article will be made available by the authors on request.

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
