# Peer review of "5-Hydroxytryptophan, a Precursor for Serotonin Synthesis, Alleviated Cognitive Dysfunction in a Mouse Model of Sepsis-Associated Encephalopathy"

_biomedicines, 2025, doi:10.3390/biomedicines13102319_

Round 1

Reviewer 1 Report

Comments and Suggestions for Authors

General impression of the manuscript:

The authors have conducted substantial research and presented novel findings that support the role of serotonergic neuromodulation in regulating cognitive functions in SAE. However, the manuscript is poorly structured, the experimental design is not clearly described, and the data presented in the figures appear contradictory. The figure legends hinder rather than facilitate understanding. Additionally, the choice of statistical analysis is not well justified.

The manuscript may be considered for publication only after major revisions have been made.

Several specific comments are also provided.

1) The description of the animal groups lacks clarity. The abundance of information presented in both the schematic diagram and Table 1 does not facilitate, but rather complicates, the understanding of the experimental design.

According to Figure 1, all six groups of animals underwent both the learning and memory phases of the Barnes maze. However, Table 1 suggests that only the LPS + 5-HTP 100 mg/kg group underwent both phases, and this group consisted of only 6 animals for the learning phase and 3 for the probe trial, which makes it non-representative.

From the table, it can be inferred that the probe trial was conducted only in this single group (LPS + 5-HTP 100 mg/kg).

The experimental design must be described more clearly, and the table should explicitly indicate in which groups animal deaths occurred.

2) Three groups of mice with a sepsis model were formed: LPS + 5-HTP 10 mg/kg, LPS + 5-HTP 25 mg/kg and LPS + 5-HTP 100 mg/kg. However, the control group is represented by only one group: Control+5-HTP10 mg/kg. Why were the corresponding controls, Control+5-HTP25 mg/kg and Control+5-HTP100 mg/kg, not performed? The reason for this needs to be explained.

3) The methodology does not specify how many days the training phase in BM took. This can only be understood from the figure.

4) The statistical analysis is based on comparisons using either the two-sample t-test for parametric data or the Mann–Whitney U test for non-parametric data.

However, a one-way ANOVA followed by post hoc tests would be more appropriate for determining statistically significant differences among multiple groups.

Additionally, a two-way ANOVA could have been employed to evaluate the impact of two independent variables—the disease model (control versus sepsis) and the dose of the tested compound—on specific behavioral outcomes.

Furthermore, Figure 2A introduces a third factor: time (training days). It is unclear how the authors performed the statistical analysis for such time-series data. Clarification of the statistical approach is necessary.

5) Figure legends should not contain interpretations or results of the study. Instead, they should be clear, concise, and descriptive, allowing the reader to navigate the presented graphs independently. The current figure legends hinder understanding and require revision for clarity and structure.

6) The data in Fig. 4c and Fig. 2c contradict each other (time spent in the target quadrant).

There are differences between the Control and LPS groups in Fig. 2c, but not in Fig. 4c. In Fig. 2c, there are no differences between the LPS and LPS + 5-HTP (10 mg/kg) groups, but there are in Fig. 4c. The authors should review how the data is presented in the figures and make sure it is consistent. These contradictions may be the result of unclear group designations and incorrectly written figure captions.

7) In section 3.2 of the results, the use of only one dose of 5-HTP (10 mg/kg) in the control group of mice is also not justified.

Reviewer 2 Report

Comments and Suggestions for Authors

This study investigates the potential of 5-hydroxytryptophan (5-HTP), a serotonin precursor, to mitigate cognitive dysfunction in a mouse model of sepsis-associated encephalopathy (SAE). The authors demonstrate that 5-HTP (10 mg/kg) improves cognitive deficits induced by lipopolysaccharide (LPS), while higher doses (25–100 mg/kg) exacerbate dysfunction and increase mortality. The cognitive benefits of 5-HTP are mediated via 5-HT1A receptors, as evidenced by reversal with the antagonist WAY100635.

Novelty: The study explores 5-HTP’s role in SAE, a clinically relevant but understudied area, highlighting its therapeutic potential.

Mechanistic Insight: The identification of 5-HT1A receptor involvement provides a pathway for 5-HTP’s effects, aligning with prior findings on fluoxetine.

Translational Gaps: The study lacks data on 5-HT levels or inflammatory markers, limiting mechanistic depth.

Behavioral Specificity: Cognitive improvements were observed only in the Barnes maze, not the novel object recognition test, suggesting task-dependent effects.

Clinical Relevance: While 5-HTP is an OTC supplement, its safety in septic patients (e.g., risk of serotonin syndrome) remains unaddressed.

Reviewer 3 Report

Comments and Suggestions for Authors

The manuscript “5-Hydroxytryptophan, A Precursor for Serotonin Synthesis, Alleviated Cognitive Dysfunction via 5-HT1A Receptor in Mouse Model of Sepsis-Associated Encephalopathy " by Zhang et al., is potentially an important study. However, following concerns need to be addressed and reconciled which could improve this manuscript.

  1. Abstract Section: No need to write about 5‑HT synthesis pathways, just write briefly role of serotonergic neurotransmitter in cognitive function instead of it.
  2. Since the authors did not conduct any receptor analyses to elucidate cognitive dysfunction via the 5-HT1A receptor and instead focused only on behavioral assessments, the manuscript title should be rewrite to better reflect the content of the study and findings.
  3. Section 2.2 Pharmacology: Authors should change the title; this is just name of chemical not pharmacology
  4. How did the authors confirm that the LPS injection successfully created the SAE mouse model? Did authors check all relevant parameters? Please include this information in the Results section
  5. Figure 1. The flow chart lacks of clarity and difficult to understand. Authors should provide clear and written explanation of the experimental design.
  6. The authors need to carefully recheck all statistical analyses to ensure accuracy and correct interpretation of results
  7. Citations are missing in all behavioral tests, Include citations .
  8. The authors should revise the conclusion to align with the presented results and findings.
  9. I suggest to the authors, read the whole manuscript very carefully there are several grammatical mistakes and many sentences are very confusing. Correct the typo error in 2.3 section Experimental Design and Drug Administration

Thanks

Round 2

Reviewer 1 Report

Comments and Suggestions for Authors

I am satisfied with the authors' responses to the comments made. The manuscript can be accepted for publication if the authors include in the discussion section the limitations in the data obtained, which they discuss in their response to question 6.

Reviewer 3 Report

Comments and Suggestions for Authors

Thanks for addressing all the comments.

Author Response

Thanks for your suggestion and guidance.